# Cyclophilin A Regulates Tripartite Motif 5 Alpha Restriction of HIV-1

**DOI:** 10.3390/ijms26020495

**Published:** 2025-01-09

**Authors:** Tingting Wang, Daniel Becker, Augustin Penda Twizerimana, Tom Luedde, Holger Gohlke, Carsten Münk

**Affiliations:** 1Clinic of Gastroenterology, Hepatology and Infectious Diseases, Medical Faculty, Heinrich Heine University Düsseldorf, 40225 Düsseldorf, Germany; tingting.wang@med.uni-duesseldorf.de (T.W.); augustin.penda@gmail.com (A.P.T.); tom.luedde@med.uni-duesseldorf.de (T.L.); 2Institute for Pharmaceutical and Medicinal Chemistry, Heinrich Heine University Düsseldorf, 40225 Düsseldorf, Germany; d.becker@hhu.de; 3Institute of Bio- and Geosciences (IBG-4: Bioinformatics), Forschungszentrum Jülich GmbH, 52425 Jülich, Germany

**Keywords:** HIV, SIV, capsid, CYPA, TRIM5, MX2, CPSF6, capsid inhibitor

## Abstract

The peptidyl-prolyl isomerase A (PPIA), also known as cyclophilin A (CYPA), is involved in multiple steps of the HIV-1 replication cycle. CYPA regulates the restriction of many host factors by interacting with the CYPA-binding loop on the HIV-1 capsid (CA) surface. TRIM5 (tripartite motif protein 5) in primates is a key species-specific restriction factor defining the HIV-1 pandemic. The incomplete adaptation of HIV-1 to humans is due to the different utilization of CYPA by pandemic and non-pandemic HIV-1. The enzymatic activity of CYPA on the viral core is likely an important reason for regulating the TRIM5 restriction activity. Thus, the HIV-1 capsid and its CYPA interaction may serve as new targets for future anti-AIDS therapeutic agents. This article will describe the species-specificity of the restriction factor TRIM5, understand the role of CYPA in regulating restriction factors in retroviral infection, and discuss important future research issues.

## 1. Introduction

Human immunodeficiency virus 1 (HIV-1), a lentivirus, is the result of successful cross-species transmissions of simian immunodeficiency viruses (SIVs) from chimpanzees and gorillas to humans [1,2]. HIV-1 is subdivided into four groups, HIV-1 M and N, which evolved from the adaptation of the SIV of the central African chimpanzee *Pan troglodytes troglodytes* (SIVcpzPtt) in humans, and HIV-1 P and O, which originated from the western lowland gorilla (*Gorilla gorilla gorilla*) lentivirus (SIVgor) [3,4] (Figure 1). In contrast, HIV-2 is the result of multiple cross-species transmissions of sooty mangabey SIV (SIVsmm) [5]. HIV-1 M is the primary cause of the pandemic of AIDS worldwide [2]. Although antiretroviral therapy (ART) has played a positive role in the prevention and treatment of HIV-1 infection, it does not cure the virus infection, and drug resistance can evolve [6,7]. As a result, novel compounds that are effective against drug-resistant HIV-1 strains and novel therapeutic classes that target unexplored viral sites must be developed. A better understanding of the intracellular interactions of HIV-1 may facilitate the identification of new treatment targets. Herein, we provide details of the HIV-1 capsid structure and CYPA-binding elements within this structure and how HIV-1 M escapes the restriction of human (hu) tripartite motif 5 alpha (TRIM5α) through CYPA binding. We then describe the restriction of huTRIM5α to HIV-1 non-M, which has been reported to be regulated by the CYPA-binding loop of the capsid. TRIM5 plays an important role in protecting cells from cross-species transmission of retroviruses, so understanding the detailed mechanism by which CYPA regulates TRIM5’s restriction of HIV-1 will help inspire the development of new methods to inhibit HIV-1 infection.

Nascent HIV-1 particles assemble at the infected cell’s plasma membrane and bud as immature virions with a membrane envelope [8]. During the initiation of virion maturation, the Gag-Pol polyprotein is hydrolyzed into reverse transcriptase (RT), integrase (IN), and protease (PR) enzymes, where the PR cleaves the Gag precursor protein into three viral structural proteins: matrix (MA), nucleocapsid (NC), and capsid (CA) proteins [9,10]. Gag cleavage results in a rearrangement of the internal structure of the virus to form the cone-shaped core from CA proteins. The MA forms a spherical shell that binds to the viral membrane and recruits the viral envelope glycoproteins (Env glycoproteins) [11,12]. Inside the MA lattice lies the viral core. The core is composed of the mature CA organized in 250 hexamers and 12 pentamers and the viral ribonucleoprotein complex consisting of two copies of the plus-stranded viral RNA genome along with NC, RT, and IN [13,14,15,16]. It has been shown that the core is the key structure of HIV-1 and plays a crucial role in almost every step during the early phase of the viral replication cycle, namely intracellular trafficking after viral fusion, nuclear import, uncoating, and integration of the viral genome into host chromatin [17,18,19] (Figure 2).

To coordinate the early stages of the HIV-1 replication cycle, the core critically depends on host factors [20,21]. Cyclophilin A (CYPA and PPIA), an example of the several interactors of the core [22,23,24,25,26], is a member of the cyclophilin family of immunophilins with ubiquitous cellular distribution [27,28,29]. Many cyclophilins possess the peptidyl/prolyl *trans*-*cis* isomerase (PPIase) activity that catalyzes the isomerization of the peptide bond upstream of proline residues in proteins [30,31]. Although this review focuses on CYPA, cyclophilin B (CYPB), another member of the cyclophilin family encoded by the human *PPIB* gene, has been shown to regulate HIV-1 nuclear import [32]. Among the eighteen different cyclophilin isoenzymes in humans, CYPA, the most abundant member of the cyclophilin family and the major player in cellular PPIase activity, is encoded by the *PPIA* gene located on chromosome 7 [33]. CYPA is an intracellular binding partner for the small-molecule immunosuppressant cyclosporin A (CsA) and participates in many biological processes, such as supporting the translation of intrinsically disordered proteins, protein folding, post-translational modifications, protein transport, the assembly of essential cellular protein complexes, and cell signaling [34,35,36]. In addition, CYPA plays a critical role in homologous recombination DNA repair following replication fork stalling, and its inhibition by CsA renders some cancer cell lines highly sensitive to cell death [37].

CYPA inhibits or promotes HIV-1 infection in a cell-specific manner and is intricately involved in many steps of the viral replication cycle, from cytoplasmic trafficking to genome integration [26,38,39,40]. The molecular mechanisms by which cyclophilins regulate HIV-1 infection are complex and not yet fully understood. Recent studies have shown that CYPA could promote the interaction of the viral core with host factors, including cleavage and polyadenylation specificity factor 6 (CPSF6) [38], SAD1 and UNC84 domain containing (SUN) 1 [41], SUN2 [42], simian TRIM5α [43,44,45,46], and myxovirus resistance 2 (MX2) [47,48,49]. CYPA has an opposite role in the context of human TRIM5α: CYPA protects HIV-1 M against human TRIM5α restriction, while the Old World monkey TRIM5α variant needs CYPA for anti-HIV-1 activity [44,45,50,51].

## 2. Capsid Structure and CYPA Binding Sites

The mature HIV-1 core is composed of around 250 capsid hexamers and 12 capsid pentamers, with seven pentamers and five pentamers distributed at both ends of the core (Figure 3A,B) [11,52]. The pentamer provides a downward inclination angle for the surface of the originally regular hexamer honeycomb, inducing the bending of the closed shell to form a unique conical capsid structure [52]. The HIV-1 capsid (CA) is a 231-residue protein that folds into two largely α-helical domains—an N-terminal domain (CA-NTD) and a C-terminal domain (CA-CTD)—connected by a short linker [53,54] (Figure 3C). The CA-NTD consists of seven α-helices and a characteristic extended CYPA-binding loop [55] (Figure 3C). The CA-CTD is made up of four α-helices, a short 3_10_ helix, and the main homology region (MHR), a highly conserved element needed for viral replication in all ortho-retroviruses, a subfamily of retroviridae that includes lentiviruses [54,56]. The NTD stands atop the CTD, facing the external environment and arranged into hexamers and pentamers, while the CTD forms the surface, facing the interior of the cone (Figure 3B). The hexameric and pentameric oligomers are formed by NTD-NTD and NTD-CTD interactions [13,57], while the CTD interacts to form homodimers and homotrimers that connect the lattice [58,59].

During infection by retroviruses such as HIV-1, many host factors bind to the viral core through several residues, particularly at the N-terminal domain [19,61,62,63,64]. CYPA has been shown to bind directly to the capsid CYPA-binding loop. Experiments involving the depletion of CYPA in cells or the inhibition of capsid binding by the use of CsA provided important information about the relevance of the interaction between CYPA and the HIV-1 capsid [39,65,66,67]. X-ray crystallography, molecular dynamics (MD) simulations, cryo-electron microscopy (cryoEM), and cryo-electron tomography (cryoET) have enabled the detailed visualization of the binding of the CA by CYPA [68]. The crystal structure of CYPA in complex with the CA-NTD shows that CA residues A88-G89-P90 and the configuration of the CYPA-binding loop are important for the binding interaction (Figure 4A–C) [55,69]. Mutations in the CA CYPA-binding loop directly affect viral replication [70].

Interestingly, cryoEM and MD simulations recently showed that CYPA recognizes not just the CA protomer but three different monomers belonging to adjacent hexamers simultaneously [71]. One of the monomers interfaces with CYPA via the classical CYPA-binding loop (Figure 4A–C), but the other two interactions take place in two binding modes at nonclassical sites: one in which CYPA binds to the dimer interface (Figure 4D), and the other in which one CYPA binds above the dimer interface and another CYPA binds above the trimer interface [71,72,73,74,75]. However, it has been reported that residue mutations at one of the secondary binding sites of CYPA (A25, K27, P29, and K30; Figure 4D) did not affect the affinity of CYPA for the CA lattice [76]. Therefore, CYPA binds to the HIV-1 capsid through both classical and nonclassical sites, with mutations at secondary binding sites not affecting its overall affinity, suggesting flexible and compensatory binding mechanisms.

There is early evidence that CYPA binding to the core of incoming viruses is responsible for their effects on infection [66,69,77]. Some studies suggest that CYPA uses its PPIase activity to catalyze the *trans-cis* isomerization of the CYPA-binding loop, especially binding to the capsid domain of dimerized Gag, to promote the depolymerization of HIV-1 capsid complexes [78,79,80]. Recent studies have demonstrated that the interaction between CYPA and the HIV-1 capsid promotes viral infection in human cells by shielding the incoming viral capsid from the host restriction factor TRIM5α [44]. Interestingly, the A88V mutation in the capsid protein plays a pivotal role in this process. This mutation alters the CYPA-mediated isomerization pattern of proline residues in the CYPA-binding loop, shifting it from *trans* to *cis*, which allows huTRIM5α to restrict the virus. Thus, the A88V residue highlights the intricate balance between viral adaptation and host restriction mechanisms [44,45,50].

## 3. Retroviral Restriction by TRIM5α

The TRIM5 gene encodes multiple protein isoforms, and TRIM5α is the splice isoform with antiretroviral activity [81]. TRIM5α is also antiviral against poxviruses and tick-borne flaviviruses, but not against mosquito-transmitted flaviviruses, and it can inhibit LINE-1 retrotransposition [82,83].

The best-studied antiviral activity of TRIM5α is its ability to effectively block retroviral infection after viral entry but before viral DNA is integrated into the host cell genome [84] (Figure 1). This retroviral restriction mode is associated with the ability of TRIM5α to recognize retroviral cores [85]. TRIM5α forms a hexagonal cage of open hexameric rings surrounding the viral core [82,83]. TRIM5α also achieves additional antiviral effects in multiple ways, including autophagy induction [86,87,88,89], the activation of cellular pathways to produce cytokines [90,91], and by triggering ubiquitin-dependent innate immune signals [91,92,93,94,95].

The TRIM5α protein has the length of approximately 500 residues with an N-terminal RING (Really Interesting New Gene) domain, a B-box, and a coiled-coil (CC) region [96,97] (Figure 5). The CC domain follows a C-terminal SPRY (also known as PRY-SPRY or B30.2) domain [23]. In TRIMCyp, the C-terminal SPRY domain is substituted with CYPA, while the remainder of the protein structure is conserved [23]. The RING domain is attached to the B-box domain by a short linker (linker 1 or L1), and the SPRY domains are connected to the CC domain by an extended linker region called linker 2 or L2 (Figure 6) [98,99]. The four domains of TRIM5α cooperate to control the restriction of HIV-1 infection. The CC region and B-box structural domains promote the formation of initial dimerization and mediate higher-order self-binding of oligomers, facilitating their aggregation into cytoplasmic bodies and the formation of hexagonal TRIM5α nets around the target viral core upon retroviral entry into the cell [82,83,100,101,102,103,104]. TRIM5α synthesizes K63-linked ubiquitin chains via the E3 ubiquitin ligase activity of its RING domain that activates mitogen-activated protein kinase 7 (MAP3K7), also known as TAK1, as well as via innate immune signaling by recruiting two E2 Ub conjugation enzymes, Ube2W and the heterodimer Ube2N/Ube2V2 [91,105,106]. Such activities of TRIM5α increase after interacting with the HIV core and act as a pattern recognition receptor (PRR) specific to the retroviral capsid lattice [88,91]. In addition, the RING domain causes self-ubiquitylation of TRIM5α via K63-linked poly-ubiquitin chains [91,105,107,108]. In addition, TRIM5α accumulates into cytoplasmic bodies via a ubiquitination-coupled liquid phase separation [109]. Whether the auto-ubiquitination of TRIM5α is relevant for HIV-1 restriction is under debate; the E3-ubiquitin ligase activity of TRIM5α, however, has a function in HIV-1 restriction [86,108,110]. The E3 ligase activity of TRIM5α is seen also after mitochondrial damage and mediates the K63-linked ubiquitination of the TANK-binding kinase 1 (TBK1) that drives the activation of TBK1 and the activation of the mitophagy machinery [111].

The SPRY domain of TRIM5α has been found to be a major determinant of species-specific retroviral restriction, and thus it is not surprising that high rates of positive selection have been found within this short domain [112,113,114,115,116,117]. To elucidate TRIM5α-mediated HIV-1 cross-species restriction, the polymorphism in the TRIM5α SPRY domain has been gaining attention. The four variable loops (V1–V4) on the surface of the SPRY domain, which have conformational flexibility, are the major determinants of capsid binding specificity [118]. The weak affinity of the SPRY domain to bind directly to individual capsid proteins is due to the fact that TRIM5α recognizes the retroviral core protein lattice rather than free capsids, and thus TRIM5α requires higher-order assembly mediated by the CC region and B-box structural domains to facilitate stable binding with cores [100,102,105,119,120]. There are no crystal structures of the SPRY-CA complex available because of the low affinity of the SPRY domain to the CA [121,122]. It has been demonstrated that SPRY binds to overlapping surfaces that are shared by CA dimers, trimers, and hexamers that involve several CA molecules, with the maximum affinity for trimers [119,121,123,124]. Other studies have reported that the two SPRY domains in the TRIM5α dimer are linked together by an α-helical segment as a whole to synergistically bind the CA, which contributes to the multivalent binding of TRIM5α to the retroviral capsid [98,119]. Although it has been described that the SPRY domains may bind to the CA in several different ways, there is still uncertainty regarding the exact 3D structure of the SPRY domain. Molecular simulations and cryo-electron tomography suggest that the SPRY domain mostly forms contacts with the CA in the CYPA-binding loop and adjacent residues 115–130, which are between helices 6 and 7, and lower contact probabilities were predicted for residues 1–15 and 40–50 [125]. Interestingly, infections with HIV-1 with capsid mutations A92E or G94D resulted in lower sensitivity to CsA treatment, which is due to the inability of huTRIM5α to bind to the mutated capsids [45]. These results further suggest that huTRIM5α may bind to or near the CYPA-binding loop. Similarly, the R82C mutation in α-helix 4 of the capsid trimer affecting the capsid surface region near the CYPA-binding loop impairs SPRY binding [124].

The main mechanism underlying the restrictive capacity of TRIM5α was long suggested to be the early disruption of the HIV-1 core and, subsequently, capsid degradation via the proteasome system [126]. The E3 ubiquitin ligase activity of the TRIM5α RING domain facilitates proteasome-dependent HIV-1 restriction. Blocking the proteasomal degradation or the ubiquitination activity of TRIM5α restores viral reverse transcription but not infection [105,110]. These data suggest that rhTRIM5α restriction involves two stages, in which rhTRIM5α first interacts with the viral core and masks its normal trafficking, followed by the proteasomal degradation of the viral core, preventing the accumulation of RT products.

In addition, TRIM5α is a selective autophagy receptor that directs the degradation of the retrovirus core through a mechanism termed “virophagy”. Studies demonstrated that TRIM5α binds to and activates important autophagy effectors, such as p62, ULK1, and BECLIN1 [89,127]. Additionally, LC3 and other ATG8 homologs—ubiquitin-like proteins conjugated to phosphatidylethanolamine that indicate autophagosomal membranes—are directly bound by TRIM5α [87]. Interestingly, autophagy is not necessary for the TRIM5α-mediated reduction in viral replication [87,128], which is consistent with findings regarding the involvement of the proteasome [110,129]. It is possible that the autophagy- or proteasome-dependent degradation processes are redundant or that cell-type-specific virus clearance mechanisms kick in after the core has been rendered inoperable. Other data suggest that the autophagy mediated by TRIM5α plays an important role in anti-retroviral signaling [88]. Independent of an HIV infection, studies have shown that for mitochondrial quality control, TRIM5α assembles higher-order structures of itself, with TBK1, and autophagy adaptors enable the autophagy-dependent clearance of damaged mitochondria (mitophagy), which is essential for preventing inflammation and cell death triggered by mitochondrial damage [111,130,131,132].

TRIM5α not only directly prevents retrovirus replication but also triggers ubiquitin-dependent innate immune signaling [88]. Considering this, TRIM5α is a PRR that identifies retroviral cores as a “pathogen-associated molecular pattern” (PAMP). Like other PRRs, TRIM5α expression levels are significantly upregulated by interferon [92,95]. The TRIM5α-mediated pattern recognition of retroviral capsids may aid in virus control during the normal course of infection in vivo [93]. In addition, TRIM5α activates both mitogen-activated protein kinase (MAPK) and NF-kB signaling pathways by cooperating with the TAK1 kinase complex, and constitutively promotes innate immune signaling, independent of its interaction with HIV-1 [91].

The TRIM5α-mediated post-entry limitation of HIV-1 is a prevalent trait in Old World monkeys, such as Rhesus macaques, but less common in New World monkeys. Although the restriction of post-entry infection with immunodeficiency viruses in New World monkey cells has not been thoroughly studied, owl monkeys are a known exception to the blockade of HIV-1 [133]. Owl monkeys lack the expression of TRIM5α but they do express TRIMCyp, a protein generated by the retrotransposition of CYPA coding sequence into *TRIM5*, replacing the SPRY domain coding sequence, which specifically shows a pattern of HIV-1 restriction not seen in other New World monkeys [133,134,135].

The human orthologue of TRIM5α (huTRIM5α) is not as effective in restricting HIV-1 as rhesus macaques TRIM5α (rhTRIM5α) when expressed ectopically, but potently limited N-tropic murine leukemia viruses (N-MLVs, a gamma retrovirus) and equine infectious anemia viruses (EIAVs, a lentivirus). Interestingly, B-tropic MLV (B-MLV) is resistant to huTRIM5α [136,137,138,139,140,141,142,143,144]. The difference in restriction sensitivity between N-MLV and B-MLV is determined by amino acid 110 in the CA [145]. Subsequently, TRIM5-like genes were also discovered in non-primate mammals, such as bovines, rabbits, mice, and pigs, and they had the characteristic of retroviral restriction factors [117,137,140,141,142,145,146,147,148]. However, the TRIM5 genes in cats and dogs were found to be truncated during evolution and lost their restrictive activity [118,137,143]. Interestingly, many of these non-primate TRIM5αs appear to have evolved from a common ancestor with antiretroviral activity [140]. Because of these collective data, TRIM5α was conceptualized as a species-specific HIV-1 restriction factor.

### 3.1. Non-Human Primate TRIM5 Proteins and HIV

Old World monkeys express TRIM5αs that have strong anti-HIV activity, e.g., rhesus macaque TRIM5α (rhTRIM5α). rhTRIM5α-mediated HIV-1 inhibition is eliminated when the B-box residue R121 is mutated in vitro, which prevents the formation of the hexagonal TRIM5 structures on the surface of the HIV-1 capsid [102,103]. Moreover, the I193A mutation within the CC domain resulted in a slight destabilization of the rhTRIM5α dimer, upon which it loses its inhibiting activity (Figure 6). Structural modeling suggests that residue I193 may be important for the correct assembling of the CC/L2/SPRY domain and that the I193A mutation may alter the position of the SPRY domain relative to the CC domain, which is associated with defective binding to the viral capsid [120]. Furthermore, mutations in the E2 binding region within the rhTRIM5α RING domain, such as R60A, abolish rhTRIM5α’s E3 ubiquitin-conjugating enzyme activity and interfere with retroviral restriction [108].

In New World monkeys, TRIM5α HIV-1-restriction activity is variable depending on the simian species. HIV-1 is inhibited by the TRIM5α proteins from woolly and spider monkeys but is not efficiently blocked from common marmoset, squirrel, or tamarin monkey TRIM5α proteins, while all these proteins inhibit SIVmac [149,150]. Owl monkeys express the exceptionally strong anti-HIV-1 variant of TRIM5α, TRIMCyp. A TRIMCyp was not only identified in the New World monkey douroucoulis or owl monkey (omTRIMCyp) but also different macaque species. In pig-tailed macaques, TRIMCyp is the only TRIM5-related allele; in contrast, rhesus macaques have alleles that express TRIM5α and, to some extent, also a separate gene for TRIMCyp [151,152,153,154]. In Owl monkeys and macaques, independent retrotransposition events with long interspersed nuclear element 1 (LINE-1) reverse transcription events resulted in the coding sequence of CYPA, replacing the SPRY domain of TRIM5α [134,135]. TRIMCyp binds to the CYPA-binding loop of the capsid, where a critical proline residue (P90) that binds the CYPA domain’s active site can be mutated in this capsid loop to essentially abolish the interaction [155]. Correspondingly, the loss of the limiting effect of omTRIMCyp can also be achieved by the mutational recognition of the key residue N369 in the CYPA domain of omTRIMCyp or the treatment of infected cells with CsA (Figure 6) [84,156,157]. However, the CYPA domain in rhTRIMCyp displays two key residues that differ from those of omTRIMCyp, with mutations that reduce restriction to HIV-1 but enhance affinity for HIV-2. The mutation of R69H binds and restricts viruses of both lentiviral lineages; the second mutation, D66N, leads to a loss of affinity for HIV-1 and produces a potent HIV-2 restriction factor [156]. The differential restriction susceptibility of rhTRIMCyp to HIV-1 and HIV-2 depends on the inherent differences in the conformation of the different lentiviral CYPA-binding capsid loops mediated by capsid residue 88 [156]. On this basis, recent studies found that the mutation of only residue H69R (NR) in rhTRIMCyp resulted in a complete loss of antiviral activity against HIV-1 N and HIV-1 O and remained inactive against HIV-1 M, while HIV-1 P and SIVgor were inhibited [50]. In contrast, the mutant N66D in rhTRIMCyp produced an antiviral protein that effectively inhibited all of the above viruses. In addition, mutations in capsid residue 88 reversed the resistance of HIV-1 M and the sensitivity of HIV-1 N, HIV-1 O, and HIV-1 P to rhTRIMCyp [50]. These data demonstrate the importance of the capsid CYPA-binding loop for rhTRIMCyp to restrict HIVs. Furthermore, unlike the SPRY structural domain, the CYPA domains are more flexibly tethered to the CC domain, and thus each TRIMCyp dimer may have a broader scope of action [99], which could account for the stronger antiviral activity of TRIMCyp against HIV-1 compared to rhTRIM5α.

Recent studies have shown that omTRIMCyp possesses similar properties to rhTRIM5α in restricting HIV-1 infection through higher-order assembly into a hexagonal lattice. However, mutations in B-box residues (R120 in TRIMCyp and R121 in rhTRIM5α (Figure 6)) significantly weaken HIV-1 restriction by rhTRIM5α but do not appear to be critical for omTRIMCyp-mediated restriction [157,158]. Moreover, in contrast to the rhesus homolog, the mutation of TRIMCyp CC-domain residue I192 (I193 in rhTRIM5α) does not abolish HIV-1 restriction [120]. These data suggest that the key residues in the B-box and CC domain determine the formation of a hexagonal lattice via the higher-order assembly of TRIM5α. The formation of the hexagonal lattice not only promotes core binding but also the effector function of the RING domain, since E3 ligase activation requires RING dimerization. Although the deletion or disruption of the B-box or CC domain results in reduced antiretroviral activity of TRIM5α, the anti-HIV-1 activity of TRIMCyp is less dependent on the integrity of the B-box and CC domain than the virus-restricting activity of the rhTRIM5α protein [158,159]. The higher affinity of the CYPA domain to the viral core compared to the lower affinity of the SPRY domain likely compensates for the higher flexibility in the B-box and CC domains of TRIMCyp.

### 3.2. Human TRIM5α and HIV-1

The TRIM5α restriction of HIV-1 in Old World monkeys is prevalent, whereas the human orthologue huTRIM5α has little effect on HIV-1 M, but inhibits non-pandemic HIV-1 N, O, and P [50,136,160]. Interestingly, approximately 4% of the Baka Pygmies from southeastern Cameroon are heterozygous for a rare variant with a stop codon in exon 8. This allele encodes TRIM5 R332X, which is truncated in the SPRY domain, does not restrict retroviral infection, but acts as a dominant negative inhibitor of huTRIM5α [161]. We speculate that such a dominant-negative human TRIM5 allele was a relevant factor for the cross-species spread of SIVs to humans and the evolution of pandemic or non-pandemic HIV-1s in Cameroon.

It was previously hypothesized that the weak restriction of HIV-1 M by huTRIM5α was determined by the V1 loop of the SPRY structural domain, and Zhang et al. verified the validity of this hypothesis by altering a single amino acid residue in the V1 loop (R437C); however, the corresponding R to C mutation in rhTRIM5α had no effect on restriction ability [162]. Many investigations have explored ways to enhance the anti-HIV-1 efficacy of human TRIM5α to improve the effectiveness of antiviral gene therapy [155,158,163,164,165]. To increase the affinity of huTRIM5α for the viral core, the amino acid at position 332 was replaced from arginine to proline (R322P), mimicking the sequence in rhTRIM5α, thereby increasing the ability of huTRIM5α to restrict HIV-1 M, albeit at a tenfold lower efficiency than rhTRIM5α [153]. In addition to P, other residues (D, E, G, H, Q, or S) at position 332 enhanced the antiviral activity of huTRIM5α against HIV-1 [164]. The huTRIM5α variants containing an R335 mutation (R335D, R335E, or R335G) similarly conferred the ability to restrict HIV-1. The R332G/R335G or R332G/R335E double mutation (Figure 6) even reduced HIV-1 infectivity by 20–50-fold [163,165]. The huTRIM5α B-box residue R119 is similar to R121 of rhTRIM5α (Figure 6) and is essential for the assembly or stability of the hexagonal lattice.

Although ineffective capsid recognition by huTRIM5α results in negligible HIV-1 restriction by huTRIM5α, an important study shows that the IFNα-mediated stimulation of the immunoproteasome activation does enable huTRIM5α to restrict HIV-1 [166]. In addition, Ribeiro et al. found that restriction by human TRIM5α is controlled by a C-type lectin receptor-dependent uptake of HIV-1, determining the protection or infection of human DC subpopulations [167]. These observations rationalize how huTRIM5α in human cells participates in the immune regulation of HIV-1 infection.

### 3.3. Non-Primate TRIM5 and Retroviruses

TRIM5α was first identified in rhesus monkeys, and, subsequently, the presence of TRIM5-like genes, or TRIMCyp genes, has also been discovered in different species of primates and other mammals. The *TRIM5* gene belongs to the paralogous cluster of *TRIM6/34/5/22* that is present in eutherians but not marsupials [114].

The bovine TRIM5α protein is closely related to the primate TRIM5α protein and has similar antiviral activity. The N-terminal RING and B-box domains in bovine TRIM5 have significant sequence identity with primate TRIM5αs, while the C-terminal SPRY domain, which determines antiviral specificity, has lower sequence identity [137,141]. Nonetheless, bovine TRIM5α can still restrict HIV-1, HIV-2, SIVmac, EIAV, feline immunodeficiency virus (FIV), and N-MLV, but not B-MLV [145]. It is worth noting that most viruses restricted by bovine TRIM5 have reduced DNA synthesis during infection, while restricted HIV-2 makes normal amounts of DNA [145]. Rabbits also carry an antiviral TRIM5 gene similar to primate TRIM5 and bovine TRIM5. The rabbit TRIM5α restricts HIV-1, HIV-2, EIAV, FIV, and B-MLV, but does not restrict SIVmac or N-MLV [140,146]. The determinant responsible for the difference in TRIM5 sensitivity in rabbits is the CA protein, which is consistent with the determinants of TRIM5α sensitivity in primates and bovine TRIM5 [168]. It has been hypothesized that these *TRIM5* genes share a common ancestor with antiretroviral properties that were later selected for by pathogenic retroviral infections, resulting in present species-specific antiviral properties [140]. In several mammalian clades, the duplication of *TRIM5* has been shown [114,116]. In rodents, there are lineage-specific gene expansions and losses in the *Trim5* cluster. Mice and rats were found to have eight and three copies of the *Trim5* genes. However, the antiviral activity of mouse or rat TRIM5-like proteins could not be demonstrated [117,142]. Interestingly, studies have shown that TRIMCyp-encoding genes exist in multiple species of the genus *Peromyscus,* commonly referred to as deer mice, which have antiviral activity against HIV-1 in a dose-dependent manner [142,169].

Felines lack the full-length TRIM5 gene. The expression of a premature stop codon 5′ to proximal exon 8 in the SPRY domain results in the truncation of feline TRIM5, which explains why feline cells do not exhibit any restriction response to retroviruses similar to that of typical TRIM5 [143]. In felines, the deletion of the SPRY domain is not replaced by CYPA to form a TRIMCyp fusion protein [143]. An artificial feline TRIM5-CYPA fusion protein (feTRIMCyp) can effectively restrict FIV and HIV-1, indicating that the feline TRIM5 RBCC domain retains antiretroviral function [144,170,171]. The discovery of the restricted function of feTRIMCyp provides a new and unique idea for HIV/AIDS infection and treatment. Similarly, in dogs, TRIM5 also lost its restrictive function due to a truncation [137]. It is the deletion of the TRIM5 gene in these species that has intrinsically provided restriction-free cell lines for most in vitro studies on the retroviral restriction of TRIM5. The species-specific restrictions displayed by TRIM5 provide a theoretical basis for improving animal models of HIV/AIDS [172].

## 4. CYPA Modulates Interactions Between Host Factors and Retroviruses

The positive effects of the CYPA-CA interactions on HIV-1 replication in human cells are reversed in some Old World monkey cells. The molecular mechanism by which CYPA affects the resistance of HIV-1 M to TRIM5α in human cells is not fully understood and may involve changes in the *trans*/*cis* isomerization activity and core dynamics of CYPA [50]. In contrast, the depletion or inactivation of CYPA relieves HIV-1 from the restriction by the rhTRIM5α protein [66,173]. Surprisingly, TRIM5α restriction-dependent CYPA-CA interaction is a common but not universal phenotype in lentiviruses. For example, most lentiviruses that are restricted by TRIM5α are regulated by a CYPA interaction with the CA. The exceptions are SIVmac and EIAV, as well as some retroviruses from non-primates, such as MLV, which are TRIM5α-restricted without a CYPA interaction [117,174,175,176]. It is worth noting that CYPA-CA binding does not only regulate the restrictive role of TRIM5α; it also affects some other restriction factors in the early stages of the viral infection [40,62,177]. As we described above (Section 3), the proposed CA interaction surface involves the CYPA-binding loop. Thus, it is discussed whether CYPA binding “blocks” the binding of huTRIM5α by introducing competition or whether the enzymatic activity of CYPA changes the conformation of the TRIM5α binding site in the CA.

### 4.1. CYPA Regulates the Anti HIV-1 Activity of TRIM5α

In Old World monkey cells, TRIM5α-mediated HIV-1 restriction is promoted through the effects of free CYPA in the target cells [51,178,179]. Similar findings were obtained if rhTRIM5α was ectopically expressed in feline or canine cells [51,173,180]. In the first experiments, HIV-1 variants with mutations in the CYPA-binding loop that prevent the binding of CYPA as G89A or P90A showed escape from rhTRIM5α restriction [51,178]. However, follow-up studies showed that HIV-1 with these mutations is restricted by rhTRIM5α similar to wild-type HIV-1 [173,180,181,182]. Interestingly, for SIVagmTAN, unlike HIV-1, CYPA does not enhance the rhTRIM5α restriction against the virus, even though the CA of SIVagmTAN does bind CYPA [182,183,184]. By exchanging regions of the CA between SIVagmTAN and HIV-1, the determinants of this phenotype were found to center on the loop between helix 4/5 (loop 4–5) and 6/7 (loop 6–7), which is the CYPA-binding loop [184]. This suggests that for some of the anti-HIV-1 activity of some TRIM5α proteins, the interaction of CYPA with other cellular components may be more critical than its direct binding to the core.

Recently, it was reported that CYPA offers protection against the restrictive effect of huTRIM5α on HIV-1 by binding to the viral core [44,45] (Figure 1). Disruption of the CA-CYPA interaction by capsid mutations (P90A or G89V in the CYPA-binding loop) or CsA treatment significantly enhanced the affinity of huTRIM5α to the HIV-1 core and the antiviral activity [45]. The reason why non-pandemic HIV-1s (N, O, and P) did not spread widely in humans is unknown, with the restricting role of huTRIM5α being one of the possible reasons [50,160]. Although the observed affinity of non-pandemic HIV-1 capsids for CYPA is similar to that of HIV-1 M, huTRIM5α exhibits significant restrictions [50] (Figure 1). Thus, the CYPA interaction with non-pandemic HIV-1s does not protect strongly against TRIM5α [50]. It has been reported that capsid residues 50 and 88 are important for TRIM5α’s capacity to restrict HIV [50,160]. Moreover, in the CYPA-binding loop, valine for HIV-1 N and HIV-1 P capsid proteins or methionine for HIV-1 O located at capsid residue 88, rather than HIV-1 M alanine, can affect the *trans*/*cis* isomerization on G89-P90 peptide in the CYPA-binding loop of the HIV-1 capsid, likely indirectly influencing binding or sensitivity to huTRIM5α [50]. In HIV-1 M, in contrast to non-pandemic HIV-1s, bound CYPA has a predicted *trans*/*cis* isomerization activity on the capsid G89-P90 peptide that likely changes the binding strength or antiviral action of huTRIM5α through allosteric ways, which supports early assumptions [80,185,186]. Based on our model, HIV-1 M affects huTRIM5α binding by producing a higher proportion of *cis* conformations in the CYPA-binding loop through a faster CYPA-mediated isomerization process. This possibility was verified by performing potential of mean force (PMF) calculations on the isomerization reaction in the presence of CYPA, which revealed that the isomerization barrier of HIV-1 M was reduced the most compared to other HIVs [50]. Accordingly, the kinetics of the *trans*/*cis* isomerization of the CYPA-binding loop mediated by CYPA enzymatic activity on the viral core may determine the sensitivity to huTRIM5α.

### 4.2. CYPA in the TRIM5α Restriction of Other Retroviruses

In addition to effectively restricting the human lentiviruses HIV-1 and HIV-2, rhTRIM5α can also restrict retroviruses from other species (Figure 7) [175]. The rhTRIM5α restriction of HIV-1 is regulated by CYPA, and the ancestor of HIV-1, SIVcpz, is subject to the same restriction mechanism [176]. In contrast, CYPA does not enhance the rhTRIM5α restriction of SIVagmTAN, although the CA of this virus, like HIV-1, does bind CYPA [66,182,184]. However, the capsid of rhesus monkey SIV (SIVmac) does not interact with CYPA, and rhTRIM5α cannot restrict the closely related SIVmac, but this does not mean that it completely escapes TRIM5α, as shown by its sensitivity to distantly related TRIM5α from the New World squirrel monkey [174]. Squirrel monkey TRIM5α blocks SIVmac infection after DNA synthesis and is not saturable with restriction-sensitive virus-like particles [174]. HIV-2, largely confined to West Africa, is less replicative, less transmissible, and less pathogenic [2,187]. HIV-2 has a weak but specific affinity for CYPA and is restricted by rhTRIM5α [156]. rhTRIM5α also effectively blocks FIV infection, but it is unknown whether its restriction is related to CYPA, to which the FIV capsid can similarly bind [182]. Finally, rhTRIM5α is restrictive to EIAV and N-MLV but not B-MLV, which are both not CYPA-regulated [112,138,188,189,190,191]. Therefore, TRIM5α dependence on CYPA-CA interactions for restriction function is a common but not universal phenotype in retrovirus restriction.

huTRIM5α has slightly different restrictions on primate retroviruses than rhTRIM5α, but is otherwise similar. SIVcpz is protected by CYPA to escape huTRIM5α, as is HIV-1 M, whereas SIVgor, the progenitor virus of HIV-1 O and HIV-1 P, is restricted by huTRIM5α, similarly to both descendant viruses. In all these viruses, residue 88 in the CYPA-binding loop plays a determining function [50] (Figure 2). HIV-2 cores show high susceptibility to huTRIM5α, with a less clear dependence on CYPA [187].

### 4.3. CYPA and Nuclear Entry Host Factors

The nuclear import of the viral core during infection depends on the CA protein. In addition to CYPA, several host factors are discussed as important players in this step, pro-viral factors such as nucleoporine proteins (NUPs) and CPSF6 or antiviral proteins such as MX2.

CPSF6 is a nuclear protein consisting of an N-terminal RNA recognition motif (RRM) domain, a middle proline-rich domain (PRD), and a C-terminal Arg/Ser-like domain (RSLD), in which the PRD mediates binding to HIV-1 CA [63,194,195,196,197,198]. CPSF6 binds to the HIV-1 capsid at the interface between the two CA monomers defined by helices 3, 4, and 5 (Figure 8), and the disruption of binding is mediated by changes in residues F321 in the PRD or by mutations in the CA, including residues 57, 70, 74, 77, and 105 [63,177,199,200,201]. The RSLD in CPSF6 is directly bound by β-karyopherin transportin 3 (TNPO3), which influences the nuclear import of the protein (Figure 2) [195,196]. Accordingly, the truncation of the C-terminus of CPSF6 (e.g., a truncation at residue 358, CPSF6-358) leads to the localization of the protein to the cytoplasm and inhibits HIV-1 infection by binding to the capsid and sequestering it from the nucleus [196,200,202,203]. HIV-1’s ancestor SIVcpzPtt shows a similar restriction mechanism to CPSF6-358, but the related SIVcpzPts is not inhibited [62]. Moreover, CPSF6 has been shown to contribute to HIV-1 trafficking to nuclear speckles. Although most established cell lines do not experience an overall drop in HIV-1 infection due to decreased CPSF6 expression or capsid binding to CPSF6, viral DNA integration is misdirected outside of gene-dense, transcriptionally active, host chromatin-to-heterochromatic, lamina-associated areas [26,177,204]. Recent data suggest that CYPA binding to the HIV-1 core prevents untimely binding of CPSF6 to the capsid in the cytoplasm. The inhibition of this binding leads to the formation of higher-order CPSF6 complexes, which disrupt HIV-1 capsid assemblies in vitro, alter capsid trafficking, and reduce infectivity [38].

The N74D HIV-1 CA mutant lost binding to CPSF6 and, interestingly, to CYPA [205]. The loss of CYPA protection in the N74D virus causes inhibition via human TRIM5α [206]. This mutation has also been identified as a resistance mutation against lenacapavir, a capsid-targeting antiviral [207]. Additionally, HIV-1 N74D capsid mutant viruses are restricted by the TRIM5-analog TRIM34 [208,209]. TRIM34-restricted lentiviral capsid activity is dependent on TRIM5α, but TRIM5α restriction is not TRIM34-dependent because TRIM34 knockdown has little effect on CYPA-binding-deficient P90A capsid mutant viruses [208,209]. This suggests that N74D impacts the “conformation” of the viral core, which leads to both a loss of CYPA binding and a newly formed interface for a TRIM34 complex.

To enter the nucleus, HIV capsid proteins make direct contact with several nucleoporine proteins. For the NUPs NUP35 and POM121, interaction with the viral core was shown to be dependent on CYPA binding to a CA [205]. The antiviral activity of the cellular MX2 protein is also regulated by the CA-CYPA interaction. MX2 is an interferon-induced GTPase consisting of three domains (the stalk domain, the GTPase domain, and the bundling signaling element (BSE) domain). Its unstructured N-terminal domain (NTD) is critical for its restriction activity [210,211]. MX2 is a strong inhibitory factor for various retroviruses, such as HIV-1, SIV, EIAV, and FIV, blocking virus replication by binding to the capsid, and in turn preventing uncoating, nuclear import of the viral core, and viral DNA integration [212,213,214,215,216]. MX2 is reported to target the HIV-1 capsid by recognizing a negatively charged pocket defined by 12 glutamate residues (three copies of E71, E75, E212, and E213) created at three hexamer intersections [211,217]. It is thought that the R11-R12-R13 sequence on the MX2 NTD binds to this negative hexamer pocket, creating electrostatic attraction between the two proteins [217]. Nucleoporins mediate the antiviral activity of MX2 in a CA-CYPA-dependent manner, and the elimination of GTPase activity alters the dependence on specific NUPs for MX2 activity [215,218]. Temporal and spatial details of the MX2-mediated HIV-1 inhibition showed that MX2 assembles phenylalanine-glycine (FG)-rich nucleoporins into cytoplasmic biomolecular condensates that likely act as nuclear pore decoys that interact with incoming capsids to trap them or induce premature viral genome release [219] (Figure 2). HIV-1 CA mutant viruses (G89V and P90A) defective in CYPA binding, shRNA-mediated CYPA depletion, or CsA treatment have been reported to result in insensitivity to MX2, suggesting that the antiviral activity of MX2 is affected by the CYPA-CA interaction [47,48,49,215,216,220]. Interestingly, HIV-1 N74D reduced its sensitivity to MX2 [218]. Another study suggests that the role of CPSF6 in nuclear imports depends on MX2, and the cooperative relationship between the two affects the NUP358-mediated nuclear import of HIV-1 cores and viral replication [214]. While these data may suggest that the nuclear import pathways are important for the MX2 restriction, the failed interaction of CYPA with the N74D virus supports an essential role of CYPA for the antiviral activity of MX2 [49].

### 4.4. CYPA and Capsid Inhibitors

Since the HIV-1 CA plays essential roles in HIV-1 replication and is highly conserved, it is a desirable target for therapeutic intervention. Recently, many novel HIV-1 capsid inhibitors have been successfully reported [221,222,223,224,225]. Most of these inhibitors can inhibit or accelerate capsid disassembly or inhibit assembly, and some of them can inhibit reverse transcription, but most of them are not currently fully approved for clinical use [226,227,228,229]. Capsid residues that are required for interactions with such inhibitors have been extensively looked at [62,229,230]. Furthermore, the influence of other host interactors of capsids on some capsid inhibitors has been highlighted [62,63,201,231,232]. These drugs can also inhibit the nuclear import and transport of the HIV-1 replication complex, possibly by competitively blocking the binding of host factors such as CPSF6 and NUP153 and/or by altering CA core stability [201,233,234].

The small-molecule capsid inhibitor PF-3450074 (PF74) has been extensively studied, and susceptibility to SIV and HIV capsids was recently reported to be modulated by CYPA, as CYPA knockout or inhibition with CsA resulted in altered inhibitor’s activity (Figure 2) [62,231,235]. The potent inhibitor lenacapavir (LEN/GS-6207) is the first and only capsid inhibitor to be approved for clinical use. Researchers are excited that the injectable LEN used as pre-exposure prophylaxis protects people for six months with each shot and thus has the potential to reduce global infection rates [236]. Molecular docking studies showed that LEN, PF74, CPSF6, and NUP153 share the same binding site located between the capsid NTD and the CTD (Figure 8) [63,201,232]. It is likely that additional host factors also bind in this pocket. A highly resistant virus with five CA mutations (Q67H, K70R, H87P, T107N, and L111I) was discovered during the first characterization of PF74 [237]. H87P is distinct from the other residues due to its location in the CYPA-binding loop, which is distal from the inhibitor binding site. The substitution of H87P reduced the sensitivity of HIV-1 to another CA inhibitor, GSK878, by 3−4-fold, similar to LEN [233]. Together, these data suggest that CYPA binding to the core indirectly modulates the antiviral activity of these capsid inhibitors.

## 5. Conclusions and Perspectives

The research on HIV-1 evolution and invasion has uncovered a dizzying array of host factors that can interact with the CA and the viral core (Figure 9). The detailed mechanisms of TRIM5α interaction with HIV are unclear. Continued research on how CYPA regulates TRIM5 activity and other molecules will contribute to our understanding of complex host-retrovirus relationships, particularly those involving species-specific replication and zoonosis.

In addition, the discovery that CYPA regulates TRIM5α to restrict poxviruses also provides new ideas for the development of antiviral drugs for diseases in different fields. CYPA plays important but ill-defined roles in various steps of the HIV-1 replication cycle, not only regulating a variety of host factors but also greatly influencing the sensitivity of capsid inhibitors, which form a complex network of relationships centered on CYPA. Therefore, disrupting CYPA-CA interactions appears to be a promising target in anti-HIV drug design.

## Figures and Tables

**Figure 1 ijms-26-00495-f001:**
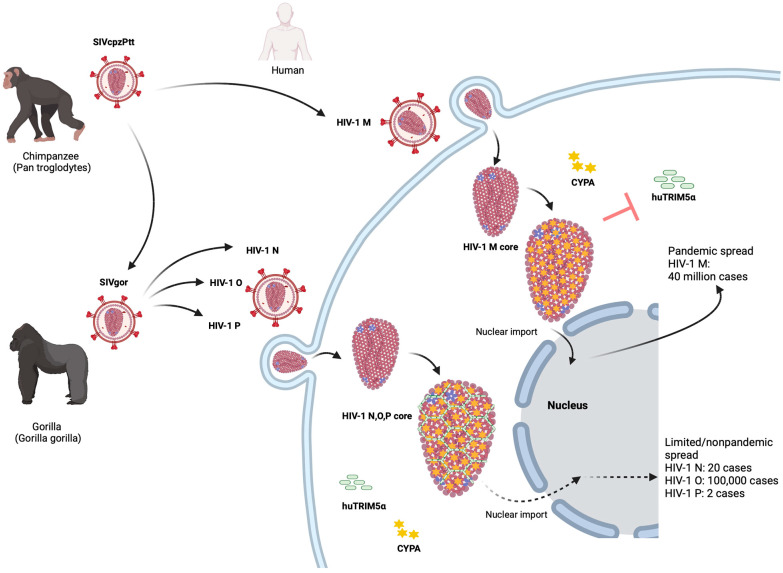
The origin and epidemiology of HIV-1. Chimpanzees acquired lentivirus infections from preyed monkeys. SIVcpzPtt from western chimpanzees was transmitted to humans and evolved into the global HIV-1 group M and the rare HIV-1 group N. SIVcpzPtt also spread to gorillas and evolved into SIVgor. SIVgor invaded humans and evolved into HIV-1 group O and rare cases of the HIV-1 group P. After HIV-1 M invades cells, CYPA binding to the capsid (CA) can protect the viral core of HIV-1 M from human TRIM5α to successfully infect host cells. TRIM5α can form a hexagonal lattice on the viral core of HIV-1, thereby inhibiting early cell infection. In contrast to HIV-1 M, the non-pandemic HIV-1 cannot use their CYPA CA interaction to protect their core against human TRIM5α.

**Figure 2 ijms-26-00495-f002:**
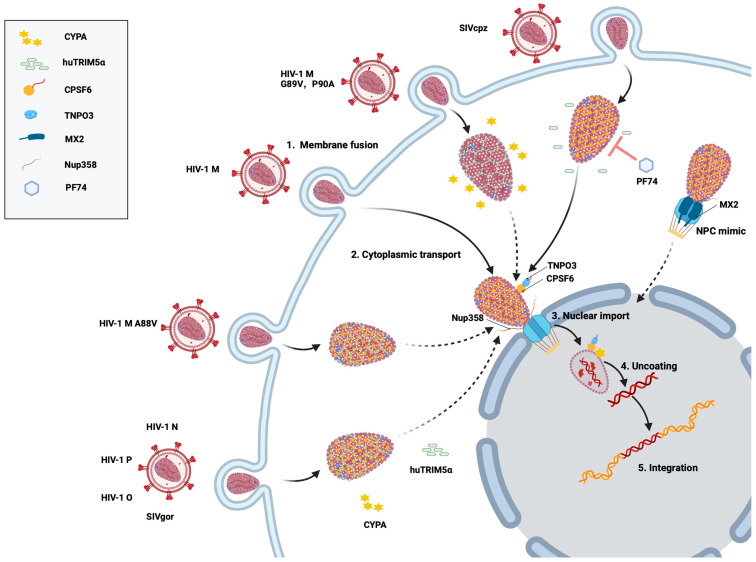
Schematic of HIV-1 M and SIVcpz cell infection. After virus–cell membrane fusion (step 1), the viral core is transported into the cytoplasm and interacts with CYPA and TRIM5α (step 2). Binding of the core with cellular proteins (e.g., CPSF6 and NUP358) promotes transport through the nuclear pore complex (NPC) (step 3). In the nucleus, after the reverse transcription of the viral RNA, the uncoating releases the viral DNA (step 4) and DNA integration can take place (step 5). SIVcpz remains mostly unaffected by the small-molecule inhibitor PF74 during invasion, whereas HIV-1 M is inhibited by PF74 under the same conditions. HIV-1 N, P, O, and SIVgor are restricted by TRIM5α. HIV-1 M capsid mutants G89V and P90A prevent CYPA from binding to the capsid, allowing huTRIM5 α to exert its restriction effect. The CA mutation HIV-1 M A88V alters the CYPA interaction with the CYPA-binding loop, shifting the isomerization pattern of its proline residues from *trans* to *cis*. This conformational change allows the recognition of the HIV-1 capsid by huTRIM5α, restoring its ability to restrict HIV-1 infection. Cytoplasmic MX2 recruits FG-Nups and importins to form biomolecular condensates that mimic nuclear pores and disrupt the nuclear targeting of HIV-1. Broken lines indicate impaired and restricted steps.

**Figure 3 ijms-26-00495-f003:**
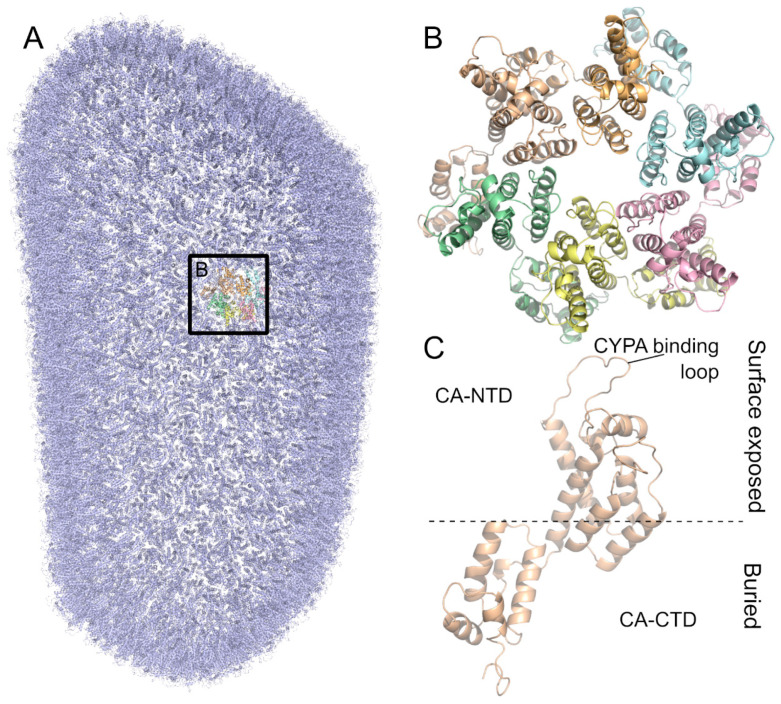
The HIV-1 capsid is arranged in a conical lattice of hexamers and pentamers of the CA protein. (**A**) Model of the full HIV-1 M capsid based on PDB ID 3J3Q [60]. All monomers are colored in blue except for one hexamer highlighted by a box. (**B**) Close-up of the HIV-1 M capsid hexamer. Monomers are colored differently. (**C**) CA protein monomer shown from the side. The CA-NTD is on the surface exposed side. The CYPA-binding loop is part of the CA-NTD. The CA-CTD is buried in the inside of the capsid.

**Figure 4 ijms-26-00495-f004:**
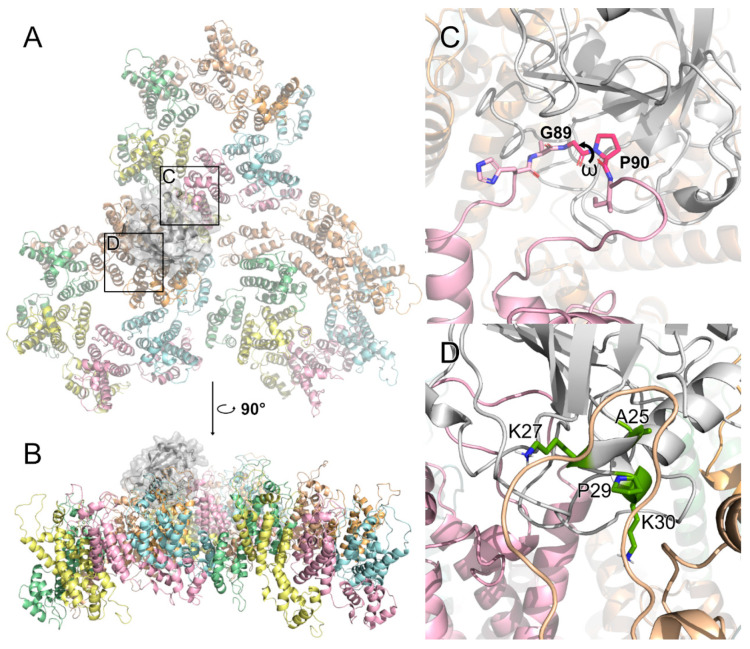
CYPA binds on the surface of the HIV-1 capsid. (**A**) The binding effect of CYPA, shown in white, on the solvent-exposed parts of CA hexamers. The region of the canonical binding site is marked with box C. The region of the putative non-canonical binding site is marked with box D. This model is based on PDB ID 6Y9X [71]. (**B**) Side view on CYPA binding on the surface. (**C**) CYPA binds to the classical CYPA-binding loop. Residues G89 and P90 are indicated, as well as the ω dihedral between them. The *trans*/*cis* isomerization of this angle is catalyzed by CYPA. (**D**) One of the secondary binding sites of CYPA to the dimer interface. Residues A25, K27, P29, and K30, deemed crucial for the interaction, are colored green.

**Figure 5 ijms-26-00495-f005:**
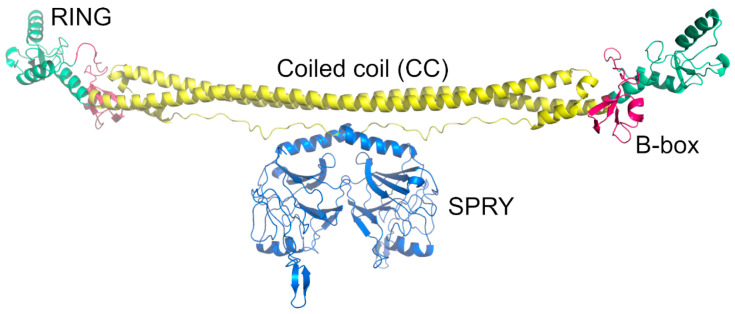
TRIM5α forms homodimers in cells. The protein consists of four domains: the SPRY domain, which forms interactions with the capsid (blue); the coiled-coil domain (yellow); the B-box (red); and the RING domain (cyan). This model was created with AlphaFold2 using two sequences as inputs.

**Figure 6 ijms-26-00495-f006:**
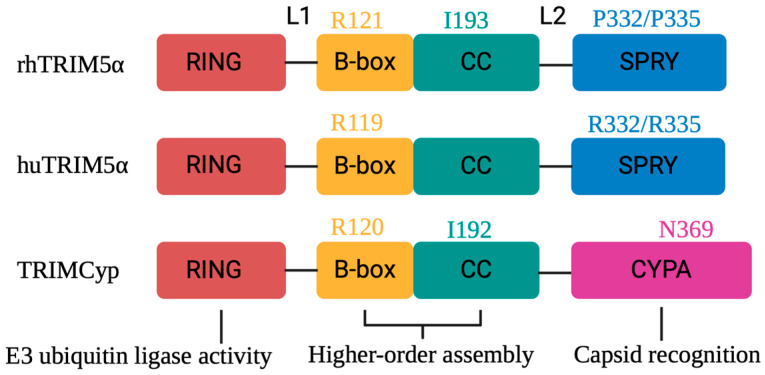
rhTRIM5α, huTRIM5α, and TRIMCyp protein domain organization and associated functions. Key amino acid residues impacting antiviral functions are depicted.

**Figure 7 ijms-26-00495-f007:**
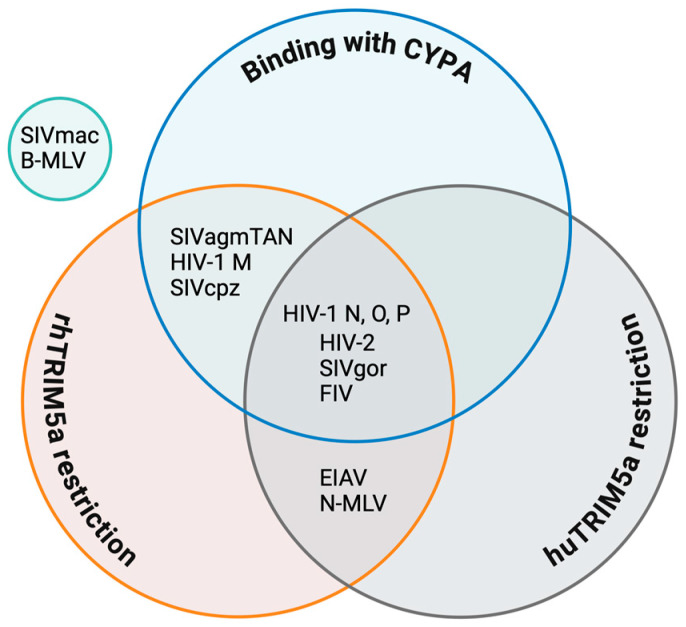
Venn diagram of the regulatory roles of TRIM5α and CYPA in HIV and other retroviruses. Data sources for individual viruses are as follows: HIV-1 M [50,175]; HIV-1 N, O, and P, which are restricted by huTRIM5α [50], as well as by rhTRIM5α, according to Twizerimana et al. (unpublished); HIV-2 [2,156,187]; SIVcpz [176]; SIVagmTAN [182,184,192]; SIVmac [139,174]; SIVgor [50]; FIV [182,193]; EIAV [138]; N-MLV [138,168]; and B-MLV [168].

**Figure 8 ijms-26-00495-f008:**
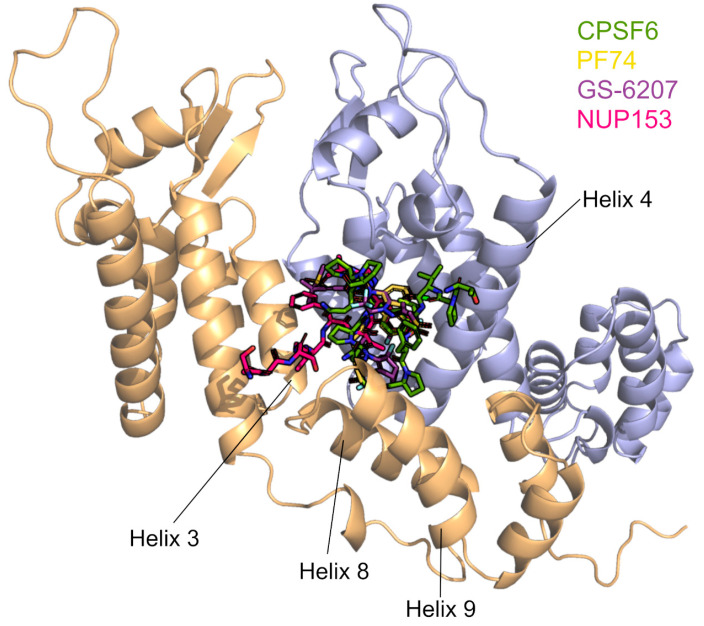
CPSF6, PF74, LEN/GS-6207, and NUP153 share a binding site in the interface of two CA monomers. The binding site is formed by Helixes 3, 4, 8, and 9. The binding positions of the four ligands are taken from PDB IDs 4WYM, 4XFZ, 6V2F, and 5TSV. The CYPA-binding loops point to the top of the figure.

**Figure 9 ijms-26-00495-f009:**
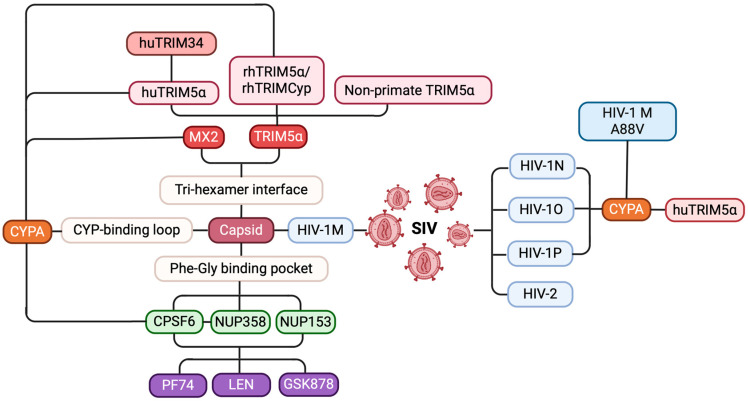
A mind map of host–virus interactions affecting HIV evolution and replication. HIV evolution and cross-species transmission (blue): SIVs evolved through cross-species transmission, leading to the emergence of HIV-1 subtypes M, N, O, and P, as well as HIV-2, with HIV-1 M being the primary strain responsible for the global pandemic. Host restriction factors and their mechanisms (red): host restriction factors such as TRIM5α and MX2 effectively inhibit HIV-1 M infection by binding to the tri-hexamer interface of the capsid. Viral cofactors that facilitate replication (green): viral cofactors like CPSF6, NUP358, and NUP153 promote viral replication by interacting with the FG pocket of the capsid. CPSF6 promotes the nuclear entry of HIV-1 pre-integration complexes by cooperating with NUP358. CYPA’s role in HIV capsid interactions (orange): CYPA protects the virus from huTRIM5α restriction by binding to the CYPA-binding loop of the capsid and modulates the antiviral activity of huTRIM34, rhTRIM5α, and MX2. Additionally, the A88V mutation in HIV-1 M alters the *trans-to-cis* isomerization of the CYPA-binding loop, restoring the inhibitory effect of huTRIM5α. Key targets for small-molecule inhibitors (purple): small-molecule inhibitors, such as PF74, LEN, and GSK878, disrupt viral replication by interfering with the interactions between the capsid and CPSF6, NUP358, and NUP153.

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
