# Peer review of "Cyclophilin A Regulates Tripartite Motif 5 Alpha Restriction of HIV-1"

_ijms, 2025, doi:10.3390/ijms26020495_

Round 1

Reviewer 1 Report

Comments and Suggestions for Authors

This is a review of cyclophilin A (CYPA) action on HIV-1 replication. It is a complete review with nice illustrations that deserves publication. I have some minor suggestions and comments.

1-since the authors mention HIV-2 several time in the text, they may want to explain the simian origin of this virus in their introduction.

2-Line 48: “non-HIV-1 M” is confusing; if it concerns non-HIV-1 then subtype M is not necessary; otherwise, HIV-1 non-M could be appropriate.

3-Lenacapavir was selected as the Breakthrough of the Year by Science Magazine. The authors may want to expand on this inhibitor.

4-Line 51: “in regulating TRIM5 to restrict HIV-1 infection” is confusing because overall CYPA does not restrict HIV-1 infection. Consider editing for clarity.

5-Line 71: the numbers of hexamers and pentamers are known and should be mentioned here for accuracy. (as in Line 122)

6-Sentence starting line 90 could be edited from a negative formulation to a positive formulation for clarity (i.e., “CYPA is crucial for homologous…”).

7-Line 86, clarifying what CsA is (e.g., a small molecule) would be useful.

8-Line 92: “killing by what”?

9-the section line 96-98 could be moved to line 83 and the authors could explain that they will focus on CYPA rather than CYPB in the rest of the document.

10-Line 100: facilitate instead of “regulate” could be used if appropriate since regulation could mean both facilitation and inhibition.

11-“restricted” should be only used for restriction factors, not for small molecules such as PF74 (line 114) – “inhibited” would be more appropriate.

12-Line 117, “The CA mutation… A88V… “ this sentence is unclear for the reader at this stage of reading the manuscript from the start and need further explanation. In addition, the effect of this mutation could be further discussed in a later paragraph, for example, that starting Line 231 or above.

13-Figure 3, add the reference for PDB ID 3J3Q in the figure legend, line 141. Same for figure 4 and model 6Y9X.

14-Line 146, this general statement should be moved earlier in the introduction.

15-Line 175: can the authors draw further conclusion about the implications of the previous statement?

16-Line 201: “or CYPA in TRIMCyp” is unclear. How does this relate to the rest of the sentence?

17-Line 284, for clarity, I suggest “with TBK1 and autophagy adaptors…”

18- Line 297 “HIV-1 interacting” is unclear.

19-The paragraph line 298 could be the start for a new #4 sub-section of the document (previous sub-section concerning TRIM5 mechanism, #3). – or moved down to paragraph/section starting line 321.

20-Speculation from the authors line 391 is intriguing. Congratulations for including this in the review.

21-Line 400: “to facilitate anti-viral activity via gene therapy”? The current formulation is confusing. Please consider editing this sentence (starting line 399).

22-Line 499, specify whether this is the affinity between capsid or core and CYPA.

23-Line 503: “TRIM5a sensitivity against HIV…”. Consider editing.

24-Information such as that provided in #4.2 is complex and dense. The reader may benefit from a table summarizing the effects described in this paragraph and possibly other places in the text (for example, interaction with CYPA, viruses restricted, …). This is for consideration by the authors and should not be viewed as a mandatory step.

25-Line 567: the authors could better define “premature”. One who assume that it means before RT is complete but please specify.

26-Line 575, the fact that N74D is a resistance mutation against lenacapavir should be mentioned.

27-Line 599: “also changes the requirement…” is unclear. Please consider editing for clarity.

Author Response

Response to Reviewers' Comments for Manuscript [Manuscript ID:ijms-3398198]
Dear Editor and Reviewers:
Thank you for reviewing our manuscript and providing constructive comments. We appreciate
your time and effort in improving our work.
We have carefully addressed all comments provided by the reviewers and revised the
manuscript accordingly. Below, we provide a point-by-point response to each comment.
Comments 1: 1-since the authors mention HIV-2 several time in the text, they may want to
explain the simian origin of this virus in their introduction.
Response 1: Thank you for pointing this out. We have added the origin of HIV-2 and the
corresponding references. --- line 39-40.
Comments 2: 2-Line 48: “non-HIV-1 M” is confusing; if it concerns non-HIV-1 then subtype M
is not necessary; otherwise, HIV-1 non-M could be appropriate.
Response 2: We have changed “non-HIV-1 M” to “HIV-1 non-M”.——line 49.
Comments 3: 3-Lenacapavir was selected as the Breakthrough of the Year by Science
Magazine. The authors may want to expand on this inhibitor.
Response 3: We added the following sentence: “Researchers are excited that the injectable
LEN used as pre-exposure prophylaxis protects people for six months with each shot and,
thus has the potential to reduce global infection rates.” ——line 675-677.
Comments 4: 4-Line 51: “in regulating TRIM5 to restrict HIV-1 infection” is confusing because
overall CYPA does not restrict HIV-1 infection. Consider editing for clarity.
Response 4: Thank you for pointing this out. We have changed the structure of the sentence
to highlight the restrictive role of TRIM5 through the regulation of CYPA.——line 51-53.
Comments 5: 5-Line 71: the numbers of hexamers and pentamers are known and should be
mentioned here for accuracy. (as in Line 122)
Response 5: Thank you for this suggestion. We have added the numbers of hexamers and
pentamers.——line 73.
Comments 6: 6-Sentence starting line 90 could be edited from a negative formulation to a
positive formulation for clarity (i.e., “CYPA is crucial for homologous…”).
Response 6: Thank you for this suggestion. We have already emphasized that “CYPA plays
a critical role in homologous ……”.——line 96.
Comments 7: 7-Line 86, clarifying what CsA is (e.g., a small molecule) would be useful.
Response 7: Thank you for pointing this out. We have added "small molecule" before CsA.—
—line 92
Comments 8: 8-Line 92: “killing by what”?
Response 8: Thank you for pointing this out. We have changed the original sentence “and
selected cancer cell lines become sensitive to killing following CYPA inhibition by CsA” to
“and its inhibition by CsA renders some cancer cell lines highly sensitive to cell death”.——
line 97-98.
Comments 9: 9-the section line 96-98 could be moved to line 83 and the authors could
explain that they will focus on CYPA rather than CYPB in the rest of the document.
Response 9: Thank you for this suggestion. We emphasized “Although this review focuses
on CYPA, cyclophilin B (CYPB)…” and moved this section to line 86-88.
Comments 10: 10-Line 100: facilitate instead of “regulate” could be used if appropriate since
regulation could mean both facilitation and inhibition.
Response 10:Agree. We have replaced “regulate” with “promote”.——line 108.
Comments 11: 11-“restricted” should be only used for restriction factors, not for small
molecules such as PF74 (line 114) – “inhibited” would be more appropriate.
Response 11: Thank you for pointing this out. We have replaced "restricted" with
"inhibited".——line 122.
Comments 12: 12-Line 117, “The CA mutation… A88V… “ this sentence is unclear for the
reader at this stage of reading the manuscript from the start and need further explanation. In
addition, the effect of this mutation could be further discussed in a later paragraph, for
example, that starting Line 231 or above.
Response 12: Thank you for this suggestion. We changed the text to: The CA mutation HIV-1
M A88V alters the CYPA interaction with the CYPA-binding loop, shifting the isomerization
pattern of its proline residues [XXX: Or singular – residue?] from trans to cis. This
conformational change allows the recognition of the HIV-1 capsid by huTRIM5α, restoring its
ability to restrict HIV-1 infection. ——line 124-127.
In addition, further discussion was given in lines 198 ff.
Comments 13: 13-Figure 3, add the reference for PDB ID 3J3Q in the figure legend, line 141.
Same for figure 4 and model 6Y9X.
Response 13: Thank you for pointing this out. We have added the reference [61] for PDB ID
3J3Q in the Figure 3, line 152 and added the reference [73] for PDB ID 6Y9X in the Figure 4,
line 173.
Comments 14: 14-Line 146, this general statement should be moved earlier in the
introduction.
Response 14: Thank you very much for your valuable suggestion. However, this sentence
serves primarily as a summary for the subsequent text, which focuses on the method of
visualizing the binding of CYPA to the CYPA-binding loop of the N-terminal domain and
highlights the key binding residues. Since the preceding text primarily discusses the structure
and function of CYPA and capsid, as well as their relationship starting from #2, we believe this
sentence is appropriately placed and does not need to be moved.——line 158 ff
Comments 15: 15-Line 175: can the authors draw further conclusion about the implications of
the previous statement?
Response 15: Thank you for this suggestion. We added further conclusion of this section
indicating flexibility and compensatory nature of CYPA binding to capsid.——line 187 ff
Comments 16: 16-Line 201: “or CYPA in TRIMCyp” is unclear. How does this relate to the
rest of the sentence?
Response 16: Thank you for pointing this out. We changed “or CYPA in TRIMCyp” to “In
TRIMCyp, the C-terminal SPRY domain is substituted with CYPA, while the remainder of the
protein structure is conserved”.——line 224 ff
Comments 17: 17-Line 284, for clarity, I suggest “with TBK1 and autophagy adaptors…”
Response 17: Thank you for this suggestion. We have changed it to “with TBK1 and
autophagy adaptors…”.——line 310.
Comments 18: 18- Line 297 “HIV-1 interacting” is unclear.
Response 18: Thank you for pointing this out. We have changed it to “its interaction
with…”.——line 323.
Comments 19: 19-The paragraph line 298 could be the start for a new #4 sub-section of the
document (previous sub-section concerning TRIM5 mechanism, #3). – or moved down to
paragraph/section starting line 321.
Response 19: Thank you for your suggestion. However, the purpose of this sentence is to
infer differences between TRIM5 in New World monkeys and Old World monkeys based on
the universal constraints of TRIM5 in Old World monkeys. Since this paragraph focuses on
New World monkeys and summarizes section #3.1, while the subsequent text delves into Old
World monkeys as a subdivision, we believe the current placement is appropriate and does
not require adjustment.
Comments 20: 20-Speculation from the authors line 391 is intriguing. Congratulations for
including this in the review.
Response 20: Thank you.
Comments 21: 21-Line 400: “to facilitate anti-viral activity via gene therapy”? The current
formulation is confusing. Please consider editing this sentence (starting line 399).
Response 21: Thank you for pointing this out. We have changed it to “Many investigations
are exploring ways to enhance the anti-HIV-1 efficacy of human TRIM5α to improve the
effectiveness of anti-viral gene therapy”.—line 426 ff
Comments 22: 22-Line 499, specify whether this is the affinity between capsid or core and
CYPA.
Response 22: Thank you for pointing this out. We have added "Capsids".—line 530.
Comments 23: 23-Line 503: “TRIM5a sensitivity against HIV…”. Consider editing.
Response 23: : Thank you for pointing this out. We have changed it to “It has been reported
that capsid residues 50 and 88 are important for TRIM5α’s capacity to restrict HIV”.——line
534.
Comments 24: 24-Information such as that provided in #4.2 is complex and dense. The
reader may benefit from a table summarizing the effects described in this paragraph and
possibly other places in the text (for example, interaction with CYPA, viruses restricted, …).
This is for consideration by the authors and should not be viewed as a mandatory step.
Response 24: We agree and make a new Figure 7.
Comments 25: 25-Line 567: the authors could better define “premature”. One who assume
that it means before RT is complete but please specify.
Response 25: Thank you for this suggestion. We further explained the topic and replaced
"premature" with “untimely”——line 605 ff.
Comments 26: 26-Line 575, the fact that N74D is a resistance mutation against lenacapavir
should be mentioned.
Response 26: Thank you for this suggestion. We have added the content that N74D is a
resistance mutation against lenacapavir and added [215] as a reference.——line 619.
Comments 27: 27-Line 599: “also changes the requirement…” is unclear. Please consider
editing for clarity.
Response 27: Thank you for pointing this out. We have changed it to “alters the dependence
on specific”.——line 643.

Reviewer 2 Report

Comments and Suggestions for Authors

In this manuscript, Wang et al provide an interesting overview on the role of cyclophilin A in the regulation of TRIMalpha restriction in HIV-1.  The interaction of HIV-1 CA with cyclophilin A is very important for understanding viral infection and pathogenesis. This is a classical research topic in HIV pathogenesis and very important as TRIM5alpha is a key host factor against HIV. The manuscript is comprehensive, but well-written and informative, containing many bibliographic entries. It is also nicely illustrated although the lack of figures/tables between pages 9 and 17 makes this part more unattractive for the non-specialist.

Adding one or two figures/tables in this part of the manuscript would probably increase the readability of the manuscript.

Otherwise, I have only a couple of minor corrections:

1)      In Figure 1, at HIV-1 O (lower left part) a space should be inserted between 100 000 and cases.

2)      In p. 10, line 313; “mice” should be used instead of “murines”

Author Response

Response to Reviewers' Comments for Manuscript [Manuscript ID:ijms-3398198]
Dear Editor and Reviewers:
We sincerely thank the reviewers for their constructive comments and valuable suggestions,
which have greatly helped us improve the quality of our manuscript.
We have carefully considered all the comments and have revised the manuscript accordingly.
Below, we provide a point-by-point response to each comment.
Comments 1:1) In Figure 1, at HIV-1 O (lower left part) a space should be inserted between
100 000 and cases.
Response 1: Thank you for pointing this out. We have added a space between 100 000 and
cases in Figure 1.
Comments 2: 2) In p. 10, line 313; “mice” should be used instead of “murines”
Response 2: We agree. We have changed “murines” to “mice”.——line 340.